# Modelling Lexical Characteristics of the Healthy Aging Population with a Natural Speech Dataset

## (Han Kunmei[1])

[1]National University of Singapore
kunmei.han@u.nus.edu

## Abstract

Modelling baseline language variation in normal aging is important for our understanding of healthy aging. Large-language databases and NLP tools enable us to conduct automated quantitative analysis of natural language data. In this study, we aim to demonstrate that using NLP tools and psycholinguistic metrics to process natural language datasets can help to set a normative benchmark of aging language. The benchmark can be applied to the assessment of cognitive aging.

## Introduction

Modelling baseline language variation in healthy aging speakers is important for understanding cognitive changes in normal aging. By setting a normative benchmark of aging language, researchers can evaluate language performance of old adults and identify potential cognitive impairments from language use. This study introduces a new way of evaluating elderly speech by adopting NLP tools and psycholinguistic metrics to natural speech data.

Old adults have been found to display reduced speech rate (Horton et al. 2010), increased pause duration (Bóna 2014), and increased lexical diversity (Moscoso del Prado Martín, 2017, Horton et al. 2010) compared to younger adults. But age-related changes in language uses are far from conclusive (cf. Cooper 1990; Luo et al. 2019). One possible reason is that the manual assessments of language abilities in previous research are too subjective, so the results can be hard to be replicated. Modern textual analysis programs such as automatic part-of-speech (PoS) tagging tools have enabled quantification of natural language with objective measures that are easy to replicate. The distribution of different PoS is one of the most used lexical measures of language samples (Chung & Pennebaker 2007, Cho et al. 2021). Besides, lexical concreteness, a well-studied psycholinguistic variable measuring the degree to which the referent of a word can be experienced by senses or actions (Paivio et al. 1968), has been used for studying semantic memory-related deficits (Breedin et al. 1994, Bonner et al. 2009).

Most exiting results about age-related language changes were gained from well-controlled experimental studies under a lab setting such as in lexical retrieval (Nicholas et al. 1985) and picture description tasks (Cho et al. 2021). These scripted or semi-structured speech data differ from naturally occurred speech. One possible reason of this research gap is the difficulty in constructing a spoken corpus before reliable speech recognition and transcribing programs were available.

In this study, we conduct a year-by-year analysis of lexical characteristics of old adults age above 60 years old with natural speech data. Since sex can also affect language uses in an aging process (Nakamura & Miyao 2008, Moscoso del Prado Martín 2017), we incorporate sex and in addition to age as main factors. We aim to demonstrate that applying NLP tools and psycholinguistic metrics to natural speech can help to set a baseline of normal aging language. The results also provide circumstantial evidence for existing findings regarding semantic memory and lexical representations.

## Method

Participants in this study are all English-speaking, cognitively healthy Singaporeans age above 60 years old. Neuropsychological battery of tests was used for the diagnosis of neurocognitive disorders and normal aging (Lim et al., 2010). Speakers age above 80 years old were excluded due to sporadic age data points. Speakers with less than 13 years of education were excluded due to the incompletion of high-school education. After exclusion, there were 243 participants in total (133 women, 110 men). The age, sex, education, and language variables were uncorrelated with each other (see Table 3 in appendices for details).

We collected about 15-min free speech from each person. Participants were asked to speak freely about their life and experiences for 15–20 minutes in English with minimal involvement from the interviewers. Speeches were recorded with audio recorders in an ordinary office setting and transcribed into text files. Then the transcriptions were then tagged with PoS information with Stanford Parser version 4.2 (Klein & Manning 2003, Qi et al. 2020). We followed the Penn Treebank tag set (Marcus et al. 1993) with adjustments for Singapore English (Lin et al. 2023, see Table 4 in appendices) and checked by trained research assistants. In total, we got 426,849 tagged word tokens. We calculated global lexical measures, including speech rate (token per minutes), lexical diversity quantified by Moving-Average Type-Token Ratio with a 20-word window (Covington & McFall 2010) and densities of major word classes (count per 100 tokens) for every individual. Lexical concreteness of major content words was calculated with the norm collected by Brysbaert and colleagues (2014). Then we conducted stepwise linear regression to explore the effects of age, sex, and education on language use with reference to adjusted R-squares. Given that the primary goal of regression is to predict the influence of age, sex, and education on lexical characteristics, rather than to explain a large proportion of language variations (which is also impossible due to the complex nature of human behavior), we will accept an R-squared if it is close to 0.1 (Ozili 2022).

## Results

Table 1 shows the textual properties of dataset. Table 2 shows the regression analysis of lexical features as a function of age and gender. The results show that PoS distribution is predominately affected by sex, while lexical concreteness is more affected by age.

| Lexical Measures | Sum | Mean (Std.) |
|---|---|---|
| Talk Time, Minutes | 3,178 | 13.08 (5.18) |
| Speech Rate, per minute | - | 137.62 (25.61) |
| All words, type | 111,722 | 459.76 (155.26) |
| All words, token | 426,849 | 1756.58 (867.52) |
| Nouns, type | 37,006 | 152.29 (63.39) |
| Nouns, token | 75,094 | 309.03 (149.17) |
| Verbs, type | 32,369 | 133.21 (50.94) |
| Verbs, token | 87,596 | 360.48 (190.46) |
| Adjectives, type | 12,847 | 52.87 (21.55) |
| Adjectives, token | 22,779 | 93.74 (46.81) |

Table 1. Textual properties of the speech dataset.

| Lexical features | | Age | | Sex | | r² |
|---|---|---|---|---|---|---|
| | | beta | *p* | beta | *p* | |
| Concreteness | Noun | 0.18 | **0.01** | -0.23 | **0.00** | 0.07 |
| | Verb | 0.03 | 0.61 | 0.05 | 0.44 | 0.00 |
| | Adj. | 0.17 | **0.01** | 0.05 | 0.46 | 0.03 |
| Density | Noun | 0.09 | 0.16 | 0.28 | **0.00** | 0.09 |
| | Verb | -0.09 | 0.14 | -0.25 | **0.00** | 0.07 |
| | Adj. | -0.01 | 0.94 | 0.19 | **0.00** | 0.03 |
| | Det. | 0.02 | 0.71 | 0.30 | **0.00** | 0.08 |
| | Prep. | -0.07 | 0.25 | 0.18 | **0.01** | 0.03 |
| | Int. | -0.14 | **0.04** | 0.10 | 0.13 | 0.02 |
| | Pron. | 0.01 | 0.88 | -0.40 | **0.00** | 0.15 |
| | Modal | -0.08 | 0.24 | 0.10 | 0.12 | 0.01 |
| Diversity | | -0.01 | 0.88 | -0.20 | **0.00** | 0.03 |

Table 2. Regression for lexical characteristics as a function of age and sex.

## Discussion

This study shows that age-related changes in lexical concreteness can be detected by large language dataset and psycholinguistics metrics. The positive association between age and lexical concreteness provides corroborative evidence for the link between concreteness and semantic memory in addition to the reduced concreteness among semantic dementia patients. The age-related changes in concreteness found in our study are different from existing findings (e.g., Cho et al. 2021), which may be due in part to the limited range of concreteness that could be achieved in semi-structured speech tasks. Natural speech provides valuable data for the assessment of cognitive abilities. Besides, we found age-related changes in concreteness differ across PoS categories. The dissociation between nouns and verbs are consistent with the findings reported in the linguistic and neuropsychological literatures (Caramazza, & Hillis 1991, Croft 2001, Vigliocco et al. 2011).

Existing text processing programs such as *Linguistic Inquiry of Word Count* (Boyd et al. 2022) rely on comparing the proportion of various word categories with embedded dictionary texts. The benchmark dictionary is decisive for the analysis results. The compilation of the dictionary texts, however, can be subjective. Incorporating widely accepted psycholinguistic metrics into annotated natural speech data reduces such subjectivity. We expect collaboration from multiple disciplines to construct such a text processing module of natural language, which can benefit clinical research and beyond.

In sum, we demonstrated that by incorporating NLP tools and psycholinguistic metrics into the construction of natural language datasets, it is possible to model language variations in healthy aging and set a benchmark for cognitive health evaluation. In the future, with a pre-processed dataset as baseline, a piece of 15-min unconstrained speech can be enough for the assessment of cognitive abilities.

**Appendices**

Table 3. Age, sex, and education correlations of participants.

|  |  | Age | Sex | Education |
|---|---|---|---|---|
| **Age** | **rho** | 1 | 0.113 | -0.079 |
|  | ***p*** | . | 0.079 | 0.222 |
| **Sex** | **rho** | 0.113 | 1 | 0.076 |
|  | ***p*** | 0.079 | . | 0.24 |
| **Education** | **rho** | -0.079 | 0.076 | 1 |
|  | ***p*** | 0.222 | 0.24 | . |

Table 4. The Penn Treebank Tag Set for Singapore English

| Categories | Tags |
|---|---|
| Nouns | NN (*book*), NNS (*books*), NNP (*Singapore*), NNPS (*Times*) |
| Verbs | VB (*be*), VBD (*were*), VBG (*being*), VBN (*been*), VBP (*are*), VBZ (*is*) |
| Adjectives | JJ (*good*), JJR (*better*), JJS (*best*) |
| Adverbs | RB (*slowly*), RBR (more), RBS (*most*) |
| Modals | MD (*can*) |
| Pronouns | PRP (*he*), PRP$ (*his*) |
| WH words | WP (*what*), WP$ (*whose*), WRB (*when*), WDT (*which*) |
| Determiners | DT (*the*), PDT (*all*) |
| Prepositions and subordinators | IN (*at*) |
| Numerals | CD (*two*) |
| Conjunctions | CC (*and*) |
| Existential *there* | EX (*there*) |
| Particles | RP (give *up*) |
| Infinitive marker | TO (*to*) |
| Sentence final particles | SFP (*lah*) |
| Local use of GOT | GOT (got) |
| Other function words | LS, POS |
| interjections | UH |
| Fragments | FRG |
| Symbols | SYM |

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
