# OpenReview forum: "Modelling Lexical Characteristics of the Healthy Aging Population with a Natural Speech Dataset"
_AAAI.org/2024/Spring_Symposium_Series/Clinical_FMs — AAAI 2024 SSS on Clinical FMs_

### Official Review · Reviewer_Zp34 · 2024-02-20
**Nice paper utilizes NLP tools and psycholinguistic metrics to analyze natural speech data**

**Rating:** 6
**Confidence:** 4

**Review:**

The paper investigates the baseline language variations in normal aging to understand cognitive changes. It utilizes NLP tools and psycholinguistic metrics to analyze natural speech data, aiming to establish a normative benchmark for aging language, which could assist in assessing cognitive aging.
Pros: The paper used NLP tools to objectively analyze natural speech data, overcoming the subjectivity in manual assessments of language abilities, besides, it provided a detailed year-by-year analysis of linguistic characteristics influenced by age and sex.
Cons: However, there are a few limitations to this study: it excludes individuals older than 80 years and those with less than 13 years of education, which could omit valuable insights from these groups. While the approach reduces subjectivity compared to manual assessments, the choice and interpretation of psycholinguistic metrics could still introduce bias.

Overall, the study presents a step forward in understanding language variation in aging.

---

### Official Review · Reviewer_aQeg · 2024-02-22
**Solid non-traditional track submission**

**Rating:** 7
**Confidence:** 2

**Review:**

This paper evaluates linguistic variation in speech data by age and gender using standard NLP tools: PoS using the Stanford Parser and the Penn Treebank tags adjusted for Singaporean English. The author goes on to derive a variety of linguistic features from this data, on which the analysis is performed. The results largely corroborate existing results related to the effect of age on linguistic variation -- particularly as it pertains to "lexical concreteness." The methodology appears to be sound, and the use of the Stanford Parser, in my opinion, constitutes foundation model usage and thus making it a relevant contribution to the workshop.

---

### Official Review · Reviewer_xRGg · 2024-02-22
**Good study focusing on the lexical characteristics of the healthy aging population**

**Rating:** 7
**Confidence:** 3

**Review:**

This paper presents a significant study focusing on the lexical characteristics of the healthy aging population using natural speech datasets and psycholinguistic metrics. The results reveal that parts of speech distribution vary with gender, while lexical concreteness correlates with age, contributing valuable information to the understanding of language variation in aging. The following points could be considered:
1. It would be beneficial to include a comparison with existing studies on younger populations or those with cognitive impairments.
2. Given that the study is based on Singaporean English speakers, how do you anticipate the findings to generalize to other English-speaking populations or languages?
3. How were the audio recordings standardized across participants to minimize environmental and technical variations?